# Association between Gut Microbiota and Metabolic Health and Obesity Status in Cats

**DOI:** 10.3390/ani14172524

**Published:** 2024-08-30

**Authors:** Kyu-Duk Yeon, Sun-Myung Kim, Jung-Hyun Kim

**Affiliations:** 1Department of Veterinary Internal Medicine, College of Veterinary Medicine, Konkuk University, Seoul 05029, Republic of Korea; gundam223@naver.com; 2KR LAB Bio Incorporation, Suwon 16229, Republic of Korea; smkim@krlab.bio

**Keywords:** cats, obesity, gut microbiome, microbiome diversity, metabolic abnormality, obesity phenotypes

## Abstract

**Simple Summary:**

In cats, obesity and metabolic disorders are significant diseases; however, the relationship between a cat’s metabolic health and gut microbiome remains unclear. This study investigated differences in gut microbiota among three groups of cats: those with normal body weight, metabolically healthy obese cats, and metabolically unhealthy obese cats. Microorganisms that metabolize carbohydrates, including *Bifidobacteriaceae*, were more abundant in obese cats. Conversely, *Ruminococcaceae* were more abundant in metabolically unhealthy obese cats. However, obesity, whether MHO or MUO, had only a minimal impact on fecal microbiota. Therefore, further studies are warranted to investigate whether gut microbiota could be a beneficial tool for the treatment or management of this condition.

**Abstract:**

Obesity is a major public health concern in both humans and animals, leading to several metabolic complications. Recent human studies have classified obesity into two phenotypes, metabolically healthy (MHO) and metabolically unhealthy (MUO) obesity based on cardiovascular and metabolic risk factors. MHO cases lack these risk factors and are protected from metabolic complications of obesity, whereas MUO cases exhibit the opposite characteristics. Moreover, recent studies have highlighted the possible role of the gut microbiome in determining metabolic health of obese individuals. However, studies on the association between the gut microbiome and obesity and metabolic abnormalities in cats are limited. Therefore, we aimed to examine the association between metabolic health phenotypes and gut microbiota composition and diversity in obese cats. We investigated hormone and serum biochemistry parameters and composition of the gut microbiota in non-obese (NO), MHO, and MUO groups. The abundances of *Bifidobacteriaceae, Coriobacteriaceae*, and *Veillonellaceae* were significantly higher in the obese versus NO group, showing a positive correlation with body mass index. The abundance of *Ruminococcaceae* was significantly higher in the MUO versus NO group, showing a positive correlation with triglyceride and total cholesterol levels. However, obesity, whether MHO or MUO, had only a minimal impact on fecal microbiota. Therefore, further studies are warranted to investigate whether gut microbiota could be a beneficial tool for the treatment or management of this condition.

## 1. Introduction

The incidence and prevalence of obesity have been increasing in humans for decades, making it a public health concern. This also holds true for cats; 59.5% and 63% of cats were reported as overweight or obese in the United States of America (USA) and New Zealand, respectively [1,2,3]. Overweight and obese cats face increased risks of various diseases, including urinary tract diseases, diabetes mellitus, respiratory and orthopedic diseases, neoplasia, dermatological and oral conditions, hypertension, and diarrhea. These diseases significantly diminish the quality of life of these animals, highlighting major welfare concerns [4].

Obesity is an important risk factor for the development of diverse metabolic abnormalities and cardiovascular diseases. In humans, these conditions can lead to a decrease in the quality of life, shortened lifespan, and concomitant economic and social burdens [5].

Recent human studies have classified obesity into two phenotypes, metabolically healthy obesity (MHO) and metabolically unhealthy obesity (MUO), based on cardiometabolic risk factors [6]. Patients with MHO have a favorable metabolic state and are protected from the metabolic complications of obesity. Although the diagnostic criteria for MHO are not universally agreed upon, most studies define MHO as a condition that meets two or fewer diagnostic criteria for metabolic syndrome, including visceral adiposity, impaired glucose tolerance, insulin resistance, dyslipidemia, and hypertension (Table 1) [7]. Categorizing obesity into MHO and MUO can help clinicians and researchers understand the mechanisms underlying obesity-related complications, develop personalized approaches for early intervention, and identify new targets for treatment and prevention [6]. Despite these benefits, studies distinguishing the MHO or MUO phenotype in veterinary medicine remain insufficient.

Cats are considered valuable animal models for studying human obesity and metabolic syndrome because the pathophysiological mechanisms underlying these diseases are similar in cats and humans. Obese cats exhibit hyperlipidemia, increased levels of pro-inflammatory cytokines, insulin resistance, and type 2 diabetes [8]. Despite efforts, the diagnosis of MUO or MHO in cats has not been firmly established, necessitating further studies. Okada et al. defined feline obesity as a condition in which cats had a body condition score (BCS) of ≥7 and met at least two of the following three criteria: low adiponectin concentration, hyperlipidemia, and high serum amyloid A (SAA) levels, similar to MUO in humans. In contrast, simple obesity was defined as having a BCS of ≥7 that did not meet the aforementioned criteria, similar to MHO in humans [9]. 

**Table 1 animals-14-02524-t001:** Clinical criteria for defining metabolic unhealthy obesity in human and cats.

In Humans (Smith et al., 2019) [7]	In Cats (Okada et al., 2019) [9]	In Cats (In This Study)
BMI > 30 kg/m^2^	BCS > 7/9	BCS > 8/9 BMI > 30 kg/m^2^
Meets the above conditions and at least two of the following:		
Waist circumference > 102 cm in men	Low adiponectin (<3 μg/mL)	Low adiponectin (<1.53 μg/mL)
Triglycerides > 150 mg/dL	Triglycerides > 165 mg/dL	Triglycerides > 165 mg/dL
HDL cholesterol < 40 mg/dL	High SAA (>200 ng/mL)	
Blood pressure > 130/85 mmHg		
Fasting glucose > 100 mg/dL		

BMI, body mass index; BCS, body condition score; HDL, high-density lipoprotein; SAA, serum amyloid A.

Unlike subcutaneous fat, visceral fat secretes adipokines (which are detrimental to metabolic health) and pro-inflammatory cytokines. Thus, a high proportion of visceral fat is a key risk factor for metabolic syndrome [10]. In humans, waist circumference correlates with visceral fat quantity [11]. However, unlike in humans, there is no physical examination metric available for visceral fat applicable to cats. However, as the amount of visceral fat in cats is inversely related to adiponectin levels, adiponectin may be an indicator of visceral fat and metabolic health in cats [12].

The possible role of the gut microbiome in determining why some individuals with obesity are metabolically healthy and others are not, is an emergent area of study in human medicine [13]. Over the past few decades, studies in humans have revealed that the gut microbiota play a major role in obesity and metabolic syndrome. Additionally, they significantly affect lipid metabolism by increasing fat synthesis and inhibiting fatty acid oxidation [14]. However, studies on the association between the gut microbiome and obesity and diabetes in cats are limited, and only a few related studies have been published [15,16,17]. Consequently, in this study, we aimed to examine the association between metabolic health phenotypes and gut microbiota composition and diversity in obese cats.

## 2. Materials and Methods

### 2.1. Study Design

Overall, 36 cats were initially recruited for this study: 26 with a BCS of 8–9/9 (considered obese, “obese group”) and 10 with a BCS of 4–5/9 (considered non-obese [NO], “NO group”) [18]. This study was conducted between May 2023 and February 2024. The study protocol was approved by the Institutional Animal Care and Use Committee of Konkuk University (Seoul, Republic of Korea) (approval number: KU23043).

All cats were client-owned animals, lived indoors, were neutered, and had an average age of 4.64 ± 0.41 years. All cats were assessed as healthy, with the exception of obesity, as defined by the American Veterinary Medical Association (AVMA) [4], based on physical examination, owner-provided medical history, and complete blood count (CBC) and serum biochemical profile assessment. There were no records of antimicrobial or anti-inflammatory medications administered during the 90-day period prior to study initiation.

### 2.2. MHO and MUO Phenotypes in Patients with Obesity

Among patients with a BCS of ≥8, those who met two of the following criteria were classified as the MUO group: triglyceride (TG) level > 165 mg/dL and adiponectin level < 1.53 μg/mL. Those who met fewer than two criteria (0–1 criteria) were classified into the MHO group (Table 1). The cutoff value of 1.53 for adiponectin was determined using the average adiponectin concentration in the NO group.

### 2.3. Dietary Data and Adaptation Period

Body weight (BW) and BCS were evaluated at the time of enrolment [18]. A 4-week dietary adaptation period assessment was conducted to ensure that all cats consumed the same commercial food (Royal Canin Indoor Adult Dry Cat Food, St. Charles, MO, USA) intended for adult cat maintenance. The diet was composed of 27.0% crude protein (minimum [min.]), 11.0% and 15.0% crude fat (min. and maximum [max.], respectively), 5.7% crude fiber (max.), and 8.0% moisture (max.). The cats were fed ad libitum. Individual maintenance energy requirements for NO and obese cats were calculated using the National Research Council equations for NO cats (100 Kcal/kg^0.67^) and obese cats (130 Kcal/kg^0.4^) [19]. At the end of the adaptation period (week 4), BW, BCS, muscle condition score, and body mass index (BMI) were measured [20]. BMI was calculated using a previously defined formula (BMI = Girth/0.70622 − Leg length/0.9156 − leg length) [21], and girth was measured at the level of the last rib [22]. 

### 2.4. Blood Sampling and Biochemical Data

Blood samples were collected at the end of the adaptation period from the external jugular or cephalic vein after 12 h of fasting. CBC and serum chemistry tests were immediately performed after blood sampling using Procyte-Dx (IDEXX Laboratories, Inc., Westbrook, ME, USA) and a Catalyst One Chemistry Analyzer (IDEXX Laboratories, Inc.). The remaining blood was cooled at 4 °C for 2 h and centrifuged at 3000 rpm for 10 min to separate the serum. The serum was frozen at −80 °C for storage for future testing. Total T4 concentration was measured using a Catalyst One Chemistry Analyzer (IDEXX Laboratories, Inc.). SAA concentrations were measured using the CAT SAA enzyme-linked immunosorbent assay (ELISA) kit (Life Diagnostics, Inc., West Chester, PA, USA). Adiponectin concentration was measured using a commercial kit (Cat Adiponectin ELISA; MyBioSource, Inc., San Diego, CA, USA).

### 2.5. Stool Collection, DNA Extraction, PCR Amplification, and Bioinformatic Data Analysis

Fecal samples were collected by the owners immediately after defecation, refrigerated at 4 °C, and transported to the hospital within 24 h. The samples were frozen at −80 °C until further analysis.

Total genomic DNA was extracted using the PureLink™ Microbiome DNA Purification Kit (Invitrogen, A29790; Carlsbad, CA, USA). Each sample DNA was eluted in a 50 μL S6 Elution buffer. The eluted DNA concentration and purity were assessed using a NanoDrop ND-1000 spectrophotometer (Thermo Fisher Scientific, Wilmington, DE, USA) and Qubit (Thermo Fisher Scientific, Wilmington, DE, USA).

DNA was extracted from canine stool samples and used as a template for PCR amplification of the V3–V4 variable regions of the bacterial 16S rRNA gene using barcoded primers containing adaptors for the Ion S5™ sequencing System (Thermo Fisher Scientific, Waltham, MA, USA). Each reaction mix contained Platinum PCR SuperMix High Fidelity (23 μL), 10 μM F Primer (1 μL), 10 μM R Primer (1 μL), and 2.5 ng/μL gDNA template (2 μL). Thermocycling conditions for the PCR amplification were as follows: 3 min at 94 °C, followed by 25 cycles of 30 s at 94 °C, 30 s at 50° C, and 30 s at 72 °C, and finally, a 5 min extension at 72°C, followed by a holding temperature of 4 °C. Following amplification, PCR reaction samples were purified using the AMPure™ XP reagent (Beckman Coulter, Inc., Brea, CA, USA) according to the manufacturer’s instructions. The PCR amplicons were quantified using a Qubit dsDNA HS Assay Kit (Invitrogen, CA, USA). After quantification, each PCR amplicon was diluted to 100 pM. Equal amounts of diluted amplicons were collected in 1.5 mL tubes to make a final volume of 50 μL.

Sequencing data were analyzed using Quantitative Insights into Microbial Ecology (QIIME) in the Thermo Fisher Ion Reporter software (version 5.18.4.0; Thermo Fisher Scientific, Waltham, MA, USA). The default settings included read length filter, ≥50; minimum alignment coverage, ≤70.0; genus cutoff, ≤97.0; species cutoff, ≤99.0; and slash ID reporting percent, ≤0.2. The read abundance filter was set at ≤2, which was not the default setting. Sequences were clustered at the operational taxonomic unit (OTU) level using Greengenes version 13.5 (Second Genome, Inc., San Francisco, CA, USA) Curated MicroSEQ 16S Reference Library version 2013.1 (Thermo Fisher Scientific, Waltham, MA, USA) and QIIME in the two databases.

To assess the diversity of bacterial species in fecal samples, alpha rarefaction curves were generated using the observed OTUs and Chao1, Shannon, and InvSimpson diversity indices, and OTU metrics were identified [23]. The bacterial communities of the control, MHO, and MUO groups were compared. Analysis of similarities (ANOSIMs) in the scikit-bio (version 0.5.8) package in Python (version 3.9.7; Python Software Foundation, Wilmington, DE, USA) was used to assess beta diversity using the Bray–Curtis dissimilarity matrix, UniFrac, and unweighted UniFrac methods. Alpha diversity metrics were not rarefied, while beta diversity was rarefied. Rarefaction was performed using the rarefy_even_depth function from the R package phyloseq (version 1.46.0; R Foundation for Statistical Computing, Vienna, Austria). In this analysis, the default value of sample.size = min(sample_sums(physeq)) was used as the rarefaction depth. The R package “ggplot” (R Foundation for Statistical Computing, Vienna, Austria) was used to visualize it using a principal coordinate analysis (PCA) plot. ANOSIMs generate an R-value between −1 and 1, with values close to 1 indicating a large difference and those close to 0 indicating no significant difference in composition between groups. The matrix used in the ANOSIMs was created using the Bray–Curtis and UniFrac methods. The Bray–Curtis dissimilarity is a statistic that measures the difference in composition between two samples. This is calculated by considering the number of microbes shared between the two samples and their relative abundances. Bray–Curtis dissimilarity takes a value between 0 and 1, where 0 means that the two samples have the same composition and 1 means that the two samples do not share any microorganisms. UniFrac is a phylogeny-based beta diversity metric that calculates similarity in microbial communities based on the evolutionary distance between microbial lineages. Weighted and unweighted UniFrac were used, with the latter only considering the presence of OTUs in the two groups being compared and the former also considering the abundance of taxa within the groups. The inclusion of weighted and unweighted beta diversity metrics in the analysis aimed to provide a comprehensive understanding of microbial community dynamics. Weighted beta diversity considers the presence and absence of taxa and their relative abundances, capturing compositional shifts and functional differences. Unweighted beta diversity focuses solely on taxonomic differences, disregarding abundance, to identify key taxa contributing to dissimilarities between samples. The OTU abundance refers to the number of reads or sequences assigned to a particular OTU in a microbial community.

### 2.6. Statistical Analyses

Summary statistics were obtained for all the parameters. Categorical variables were assessed for distributional homogeneity between groups using the chi-square test. The distribution of values was visually inspected using normal probability plots and histograms and quantitatively assessed using the Shapiro–Wilk test. For normally distributed variables, *p* values were adjusted by analysis of variance, and all pairwise comparisons were performed using Tukey’s post hoc test. Non-normally distributed variables were analyzed using the Kruskal–Wallis H test, and pairwise comparisons were performed using the Mann–Whitney U test. Initial and graphical analyses were performed using the Statistical Package for the Social Sciences (IBM SPSS Statistics for Windows, Version 23.0. IBM, Armonk, NY, USA). For all analyses, *p* < 0.05 was considered significant. Summary statistics for normally distributed values are reported as means ± standard deviations and for non-normally distributed data as medians and ranges.

## 3. Results

### 3.1. Cat Characteristics and Grouping

Of the 36 cats initially recruited, 5 were excluded during the dietary adaptation period: 3 due to food refusal, 1 due to an allergy, and 1 due to respiratory symptoms. The remaining 31 cats successfully adapted to the diet without any illness. Weight changes were within 5% of baseline values over the 4-week period. 

Thus, the final study population comprised 10 NO (BCS, 4–5/9) and 21 obese (BCS, ≥8/9) cats. The NO group included 7 males and 3 females. The MHO group had 10 males and 2 females, and the MUO group comprised 5 males and 4 females. Of the 31 cats, 29 were Korean Domestic Shorthairs, and 2 were Persians.

At the end of week 4, BW, BMI, girth, and BCS were measured (Table 2). They were higher in the MHO and MUO groups than in the NO group. No significant differences were observed between the MHO and MUO groups.

### 3.2. Blood Biochemical Data

No significant abnormalities were found in CBC, serum chemistry, or total T4 test results. Blood test results are provided in Appendix A.

TG levels were significantly higher in the MUO group (279.7 (169–375) mg/dL) than in the NO (93 (56–188) mg/dL) and MHO (107.4 (60–272) mg/dL) groups (*p* < 0.05). Adiponectin levels were 0.9 ± 0.5 μg/mL, 1.5 ± 0.3 μg/mL, and 1.5 ± 0.8 μg/mL in the MUO, NO, and MHO groups, respectively, with no statistically significant difference among the groups (*p* = 0.07). The mean values showed a decreasing trend in the MUO versus NO and MHO groups (Table 3).

### 3.3. Fecal Microbiota Analysis

In total, 1,800,083 reads passed all filters, with an average of 56,253 reads per sample (range, 37,526–90,751). Reads were mapped to two databases, Greengenes and Curated MicroSEQ 16S Reference Library, using QIIME.

### 3.4. Relative Abundance

The median relative bacterial abundance was calculated for all taxa. The most abundant phyla in all groups were (in descending order): Firmicutes, Actinobacteria, Bacteroidetes, and Proteobacteria. The most abundant families in all groups were (in descending order): *Lachnospiraceae*, *Clostridiaceae*, *Peptostreptococcaceae*, *Bifidobacteriaceae*, *Ruminococcaceae*, and *Coriobacteriaceae* (Figure 1). The most abundant genera in all groups were (in descending order): *Clostridium*, *Blautia*, *Ruminococcus*, *Bifidobacterium*, *Collinsella*, *Eubacterium*, *Dorea*, and *Peptoclostridium*.

After Benjamini–Hochberg adjustment, statistically significant differences in microbial composition were observed at the phylum, class, order, family, and genus levels (Table 4).

Abundance of Firmicutes was significantly higher in the NO group than in the MHO group (false discovery rate [FDR] < 0.05), and a similar trend was confirmed in comparison with the MUO group (*p* < 0.05, FDR = 0.054). Abundance of Actinobacteria, Actinomycetes, Bifidobacteriales, and *Bifidobacterium* was significantly lower in the NO group than in the MHO and MUO groups (FDR < 0.05). *Bifidobacterium* belongs to the *Bifidobacteriacea* family and *actinomycetes* to actinobacteria. Abundance of Coriobacteriia, Coriobacteriales, and *Coriobacteriaceae* was significantly lower in the NO group than in the MHO (FDR < 0.05) and MUO (*p* < 0.05, FDR > 0.05) groups. *Coriobacteriaceae* family belongs to the Coriobacteriales class and in turn, to Coriobacteria. Negativicutes and *Veillonellaceae* were also less abundant in the NO group than in the MHO group (*p* < 0.05, FDR = 0.14). *Veillonellaceae* belong to the Negativicutes class.

Clostridia and Eubacteriales were significantly more abundant in the NO group than in the MHO group (FDR < 0.05); this trend was also confirmed in the MUO group (*p* < 0.05, FDR > 0.05). *Peptococcaceae*, belonging to Eubacteriales, were more abundant in the NO group than in the MHO group (*p* < 0.05, FDR > 0.05).

The abundance of *Ruminococcaceae* was significantly higher in the MUO group than in the NO group (FDR < 0.05). When the MUO and MHO groups were compared, the MUO group showed a higher trend of abundance than did the MHO group (*p* < 0.05, FDR > 0.05).

### 3.5. Alpha and Beta Diversity Indices

The alpha diversity matrices did not show significant differences between groups (Figure 2). Bray–Curtis and unweighted and weighted UniFrac indices were used to assess beta diversity. No significant differences were observed between the groups (*p* > 0.05 for all comparisons) (Table 5). Moreover, PCA did not reveal clear clustering of samples according to group (Figure 3) (*p* > 0.05 for all comparisons).

### 3.6. Principal Component Analysis of Fecal Bacteria and BW, BMI, Adiponectin, TG, TChol, and fSAA Levels

A PCA of the BMI, adiponectin, TG, total cholesterol (TChol), and feline SAA (fSAA) levels resulted in distinct clusters. One cluster included BMI and *Bifidobacteriaceae*, *Coriobacteriaceae*, and *Veillonellaceae*. The second cluster comprised TG, TChol, and total bilirubin (TBIL) levels and *Ruminococcaceae*. Additionally, BMI was inversely related to *Peptococcaceae* and *Peptostreptococcaceae* (Figure 4).

## 4. Discussion

Our study found that *Bifidobacteriaceae*, *Coriobacteriaceae*, and *Veillonellaceae* species were enriched exclusively in the obesity groups (MHO and MUO), showing a positive correlation with BMI. Moreover, the relative abundance of *Ruminococcaceae* species was higher in the MUO group than in the NO group, and it was positively correlated with levels of TG, Tchol, and TBIL. However, obesity, whether in healthy (MHO) or unhealthy (MUO) groups, had only a minor impact on fecal microbiota. 

In this study, we used two criteria to distinguish between the MHO and MUO phenotypes: adiponectin levels and hypertriglyceridemia. The adiponectin cutoff value of 1.53 μg/mL was newly adopted as a diagnostic criterion for feline MUO, based on the average adiponectin concentration in the NO group.

Okada et al. proposed fSAA to be one of the diagnostic criteria for feline obesity, based on the fact that obesity causes mild inflammation. However, in this study, the fSAA was not included as a diagnostic criterion for MUO. This is because fSAA can be sensitively elevated depending on the presence of other comorbidities, which can reduce diagnostic specificity when used for the diagnosis of metabolic diseases. 

Actinobacteria, Actinomycetes, Bifidobacteriales, *Bifidobacteriaceae*, and *Bifidobacterium* were significantly more abundant in the MHO and MUO groups than in the NO group. The finding that *Bifidobacterium* levels were elevated in obese patients was contradictory to the results of previous human studies [24] but was consistent with those of previous cat studies [15,16]. In human and rodent studies, *Bifidobacterium* is a beneficial bacterium for the prevention of obesity and metabolic diseases, as it reduces blood sugar levels, induces insulin secretion, and improves endotoxemia [25,26]. Thus, it is actively being studied as a probiotic in companion animals and humans [27]. It is presumed that these differences may arise due to the fact that cats are carnivores. Further research is needed to determine the association between *Bifidobacterium* and feline obesity. 

*Bifidobacterium*, *Coriobacteriaceae*, and *Veillonellaceae* metabolize carbohydrates to produce lactate and acetate. It is possible that acetate produced by these bacteria contributes to obesity by stimulating appetite. In rodent studies, acetate produced by the gut microbiota has induced the secretion of the appetite-stimulating hormone ghrelin [28]. A study in kittens found that increases in these three bacterial groups were associated with increased acetate production from carbohydrates and increased blood ghrelin levels [29]. Further research is needed to determine why these microorganisms are more prevalent in the obese group. 

In contrast to the three aforementioned families, *Peptococcaceae* was less abundant in the obese group. Also, *Peptococcaceae* and *Peptostreptococcaceae* showed a negative correlation with BMI in the PCA plot. Members of *Peptococcaceae* have been observed in the feces of healthy cats [30,31]. Conversely, a decline in the abundance of *Peptococcaceae* is associated with feline diabetes and diarrhea [30,31]. Moreover, the abundance of *Peptostreptococcaceae* was lower in the obese group than in the NO group in a study on obese cats [32]. 

In this study, the abundance of *Ruminococcaceae* was significantly higher in the MUO group than in the NO group. In the PCA plot, *Ruminococcaceae* formed clusters with TG, TChol, and TBIL values. 

Nonetheless, reports on the role of *Ruminococcaceae* in feline obesity are conflicting. A study on cats with diabetes exhibited fewer unknown *Ruminococcaceae* than that in NO cats [15]. However, a study on kittens showed a positive correlation between TG levels and *Ruminococcaceae* in the PCA plot, similar to the results of this study [29].

Similar to the findings in cats, the effect of *Ruminococcaceae* on obesity in humans remains controversial. *Ruminococcaceae* are associated with obesity in some studies but with leanness in others [33]. Moreover, although some species of *Ruminococcaceae* (e.g., *Ruminococcus bromii*) have shown potentially beneficial effects through the breakdown of resistant starch when forming the gut microbiota, other species have been associated with human colorectal diseases, immune-mediated diseases, obesity, diabetes, and cancer [34,35]. Specifically, Kurilshikov et al. showed that *Ruminococcus* sp_5_1_39BFAA was positively correlated with liver fat content and atherosclerotic plaques in obesity [36]. Also, Zhao et al. demonstrated that *Ruminococcus* sp. N15.MGS-5 was associated with increased TG levels, decreased high-density lipoprotein levels, and progression of endometrial cancer [35].

Some *Ruminococcus* species also negatively affect metabolic health because of the effects of the SCFAs. SCFAs play a positive role in food intake and BW. However, certain SCFAs have a negative effect on obesity [37,38]. Further research is needed to determine the association between *Ruminococcaceae* and feline obesity.

Nevertheless, this study has few limitations. First, the sample size was limited and could be further increased and balanced between the control and obesity groups. Second, other potential factors that could influence the gut microbiota, such as previous dietary history, environmental differences, age, and breed, were not considered or controlled. Future studies should aim to control or standardize these variables to enhance the reliability of the findings. Third, a cat ELISA kit was used for adiponectin, which has not been used in previous studies. Ideally, the adiponectin value of 3 μg/mL, as proposed by Okada et al. in their previous research, should have been used. This discrepancy arises due to variations in reference values between ELISA kits. While the study by Okada et al. used a mouse-derived adiponectin ELISA, this study utilized a cat-derived adiponectin ELISA. Additional body fat measurements using computed tomography or dual-energy X-ray absorptiometry would be beneficial to more clearly ascertain the association between the measured adiponectin values and visceral fat content. Fourth, the owners collected, stored, and transported samples in a refrigerated environment at 4 °C. Previous research has shown that microbial composition in samples refrigerated at 4 °C for 24 h did not differ significantly from that in samples stored at −80 °C, while samples stored at room temperature showed changes in microbial composition [39]. If refrigeration was not maintained when the owners handled the samples, this could have potentially caused changes in the microbiota. For future studies, it is recommended to use kits that minimize the impact of storage and transportation on samples. Fifth, only the identification of gut microbiota was performed, with a limited number of serum metabolites measured. No fecal metabolite analysis was conducted. Future studies should include the measurement of fecal metabolites and a more comprehensive analysis of a broader range of serum metabolites. Finally, since the animals were grouped as either lean or obese, it was not possible to assess or measure risk factors for weight gain. To investigate the causal relationship between microbiota and obesity, a longitudinal study is necessary.

## 5. Conclusions

This is the first study to examine the differences in gut microbiota among cats with MHO and MUO and NO cats based on BW and adiponectin and TG levels. We observed a higher proportion of carbohydrate-metabolizing microbes in the obese cats and identified microbial components associated with metabolic abnormalities in obese cats. However, obesity, whether classified as MHO or MUO, had only a minimal impact on fecal microbiota. Therefore, further studies are warranted to investigate whether gut microbiota could be a beneficial tool for the treatment or management of this condition.

## Figures and Tables

**Figure 1 animals-14-02524-f001:**
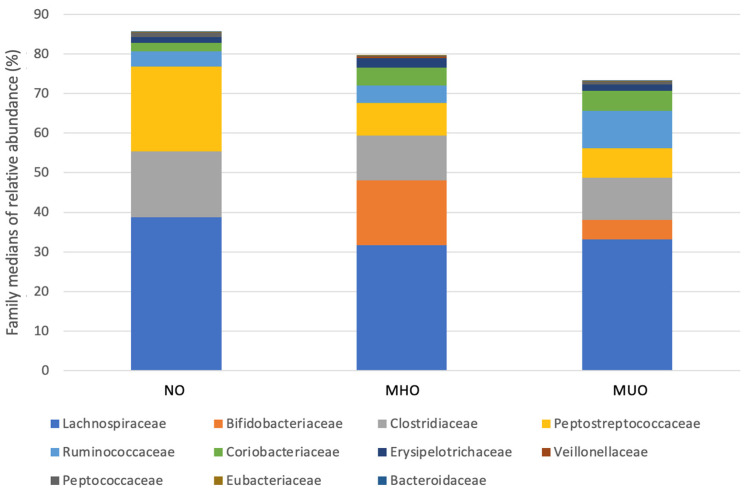
Relative abundances of predominant family originating from fecal samples of the non-obese (NO), metabolically healthy obesity (MHO), and metabolically unhealthy obesity (MUO) groups.

**Figure 2 animals-14-02524-f002:**
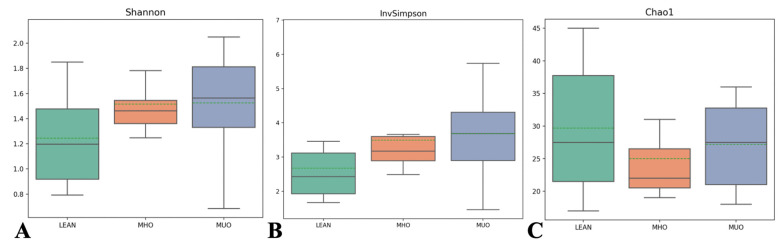
Bacterial population evenness (Shannon), diversity (InvSimpson), and richness (Chao1) in healthy, non-obese (NO) (*n* = 10), metabolically healthy obesity (MHO) (*n* = 12), and metabolically unhealthy obesity (MUO) (*n* = 8) groups.

**Figure 3 animals-14-02524-f003:**
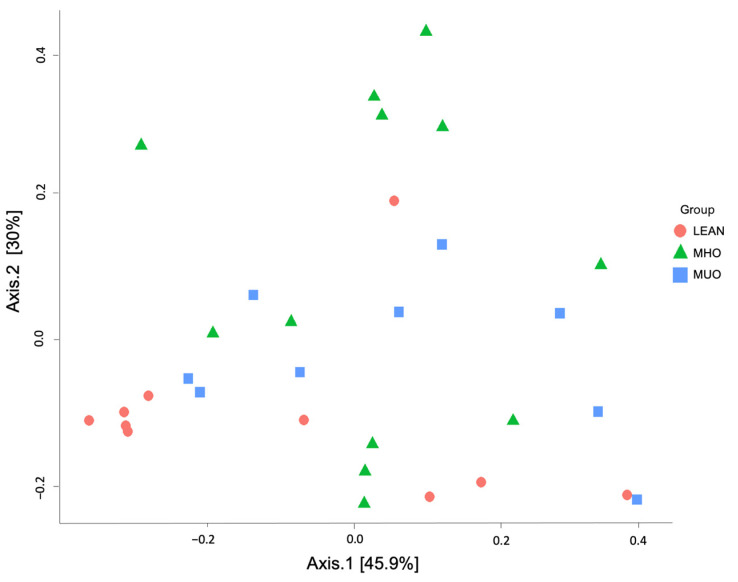
Three-dimensional principal coordinate analysis of population of the fecal microbiota of healthy, non-obese (NO) (*n* = 10), metabolically healthy obesity (MHO) (*n* = 12), and metabolically unhealthy obesity (MUO) (*n* = 8) groups. Each group is represented with a different color (see legend in figure).

**Figure 4 animals-14-02524-f004:**
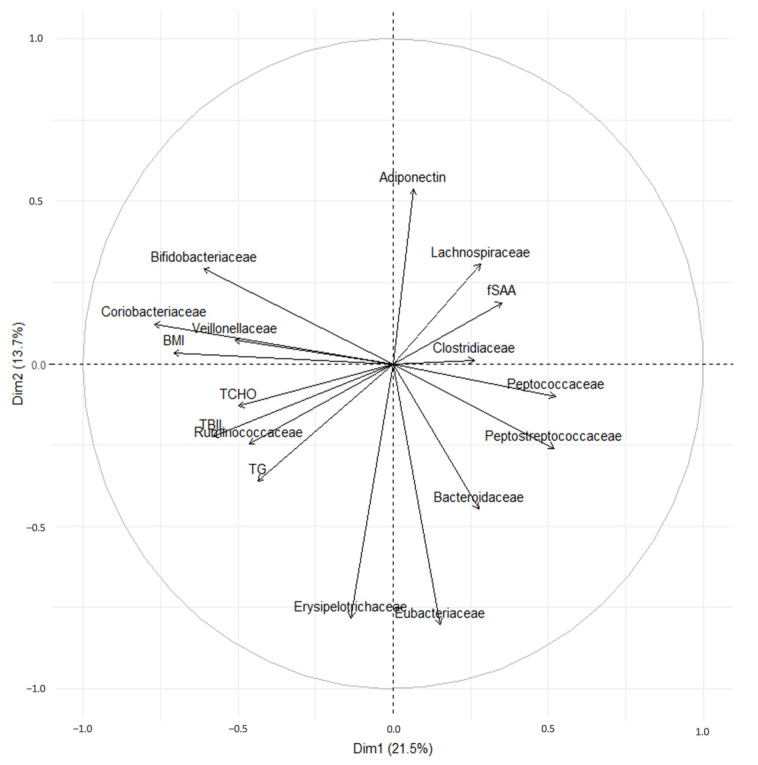
Principal component analysis loading plot of the primary fecal bacterial families and body mass index (BMI), total bilirubin (TBIL), triglycerides (TGs), total cholesterol (TChol), adiponectin, and feline serum amyloid A (fSAA) levels of interest in healthy, non-obese (NO) (*n* = 10), metabolically healthy obesity (MHO) (*n* = 12), and metabolically unhealthy obesity (MUO) (*n* = 8) groups.

**Table 2 animals-14-02524-t002:** Physical characteristics of the cats.

	NO	MHO	MUO
N	10	12	9
Sex			
Castrated male	7	10	5
Spayed female	3	2	4
Age	4 (2–9)	5 (2–9)	5 (3–9)
BW (kg)	5.0 ± 0.76 ^b^	8.4 ± 1.40 ^a^	7.3 ± 0.96 ^a^
BMI (kg/m^2^)	19.5 ± 3.93 ^b^	37.3 ± 4.68 ^a^	39.1 ± 4.79 ^a^
Girth (cm)	37.7 ± 1.95 ^b^	49.0 ± 3.52 ^a^	51.2 ± 4.89 ^a^
BCS (1–9/9)	4 (4–5)	9 (8–9)	9 (8–9)

Non-normally distributed data are provided as medians (ranges); normally distributed data are provided as means ± standard deviations. Values within the same variable with different superscripts ^a,b^ indicate statistically significant differences (*p* < 0.05) between groups. NO, non-obese; MHO, metabolically healthy obesity; MUO, metabolically unhealthy obesity; N, number; BW, body weight; BMI, body mass index; BCS, body condition score.

**Table 3 animals-14-02524-t003:** TG, adiponectin, and fSAA concentrations.

	NO	MHO	MUO
TG (mg/dL)	93 (56–188) ^a,b^	107.4 (60–272) ^a^	279.7 (169–375) ^b^
Adiponectin (μg/mL)	1.5 ± 0.3	1.5 ± 0.8	0.9 ± 0.5
fSAA (ng/mL)	226.8 (7.1–805.4)	128.2 (1.6–465.2)	111.57 (0.0–712.6)

Non-normally distributed data are provided as medians (ranges); normally distributed data are provided as means ± standard deviations. ^a,b^ Within a column, values with the same superscript letter are significantly (*p* < 0.05) different. NO, non-obese; MHO, metabolically healthy obesity; MUO, metabolically unhealthy obesity; TG, triglycerides; fSAA, feline serum amyloid A.

**Table 4 animals-14-02524-t004:** Differences in rarefied relative abundance of the most common gut microbiota at phylum to genus level among non-obese (*n* = 10), metabolically healthy obesity (*n* = 12), and metabolically unhealthy obesity (*n* = 9) groups.

	NO vs. MHO				NO vs. MUO				MHO vs. MUO			
	E	SE	*p*	FDR	E	SE	*p*	FDR	E	SE	*p*	FDR
Phylum (4/7)	**−1.320**	**3.639**	**0.002**	**0.005**	**−0.997**	**2.025**	**0.006**	**0.022**	0.750	3.349	0.256	0.341
Actinobacteria	**1.200**	**3.565**	**0.002**	**0.005**	0.778	2.005	0.027	0.055	−0.753	3.321	0.227	0.341
Firmicutes	0.482	0.358	0.598	0.797	0.374	0.419	0.935	0.935	−0.235	0.104	0.256	0.341
Bacteroidetes	0.375	0.187	0.920	0.920	0.455	0.211	0.901	0.935	0.310	0.060	0.914	0.914
Proteobacteria												
Class (7/19)	**−1.168**	**2.831**	**0.002**	**0.013**	**−0.660**	**1.458**	**0.003**	**0.023**	0.811	2.702	0.227	0.501
Actinomycetes	−0.442	1.188	0.198	0.278	−0.697	0.540	0.236	0.413	0.144	1.301	0.915	0.915
Bacilli	0.484	0.355	0.468	0.468	0.373	0.416	1.000	1.000	−0.242	0.104	0.227	0.501
Bacteroidia	**1.295**	**4.310**	**0.004**	**0.013**	1.061	2.652	0.027	0.079	−0.624	4.018	0.286	0.501
Clostridia	**−1.125**	**1.168**	**0.008**	**0.019**	−1.057	0.948	0.034	0.079	0.260	1.252	0.722	0.915
Coriobacteriia	−0.491	0.673	0.468	0.468	−0.495	1.394	0.683	0.956	−0.231	1.351	0.887	0.915
Erysipelotrichia	−0.615	0.282	0.070	0.123	0.191	0.047	0.825	0.963	0.633	0.295	0.036	0.250
Negativicutes												
Order (7/38)	0.484	0.355	0.468	0.468	0.373	0.416	1.000	1.000	−0.242	0.104	0.227	0.501
Bacteroidales	**−1.169**	**2.829**	**0.002**	**0.011**	**−0.661**	**1.458**	**0.003**	**0.023**	0.811	2.700	0.227	0.501
Bifidobacteriales	**−1.125**	**1.168**	**0.008**	**0.019**	−1.057	0.948	0.034	0.079	0.260	1.252	0.722	0.915
Coriobacteriales	−0.491	0.673	0.468	0.468	−0.495	1.394	0.683	0.836	−0.231	1.351	0.887	0.915
Erysipelotrichales	**1.295**	**4.310**	**0.004**	**0.013**	1.061	2.652	0.027	0.079	−0.624	4.018	0.286	0.501
Eubacteriales	−0.443	1.188	0.175	0.245	−0.698	0.540	0.204	0.356	0.144	1.301	0.915	0.915
Lactobacillales	−0.640	0.283	0.044	0.078	0.053	0.043	0.716	0.836	0.635	0.295	0.023	0.164
Veillonellales												
Family (11/72)	0.743	0.118	0.323	0.435	−0.049	−0.215	0.806	0.806	−0.589	0.166	0.831	0.887
*Eubacteriaceae*	−0.640	0.283	0.044	0.128	0.053	0.043	0.716	0.806	0.635	0.295	0.023	0.216
*Veillonellaceae*	−0.491	0.673	0.468	0.515	−0.495	1.394	0.683	0.806	−0.231	1.351	0.887	0.887
*Erysipelotrichaceae*	0.055	1.932	0.356	0.435	0.376	1.559	0.288	0.453	0.204	2.168	0.887	0.887
*Clostridiaceae*	0.574	4.509	0.187	0.385	0.309	4.925	0.253	0.453	−0.250	4.437	0.722	0.887
*Lachnospiraceae*	−0.424	1.108	0.692	0.692	**−1.824**	**1.062**	**0.002**	**0.018**	−0.758	1.224	0.039	0.216
*Ruminococcaceae*	0.827	2.968	0.210	0.385	0.470	3.639	0.288	0.453	−0.299	2.443	0.887	0.887
*Peptostreptococcaceae*	0.505	0.263	0.355	0.435	0.344	0.312	0.806	0.806	−0.341	0.092	0.135	0.497
*Bacteroidaceae*	**−1.125**	**1.168**	**0.008**	**0.046**	−1.057	0.948	0.034	0.124	0.260	1.252	0.722	0.887
*Coriobacteriaceae*	0.941	0.157	0.047	0.128	0.598	0.170	0.161	0.444	−0.331	0.132	0.609	0.887
*Peptococcaceae*	**−1.169**	**2.829**	**0.002**	**0.017**	**−0.661**	**1.458**	**0.003**	**0.018**	0.811	2.700	0.227	0.624
*Bifidobacteriaceae*												
Genus (16/88)	0.479	0.314	0.320	0.565	0.296	0.381	0.870	0.928	−0.308	0.142	0.239	0.893
*Bacteroides*	**−1.164**	**3.191**	**0.001**	**0.020**	−0.668	1.619	0.003	0.053	0.811	3.040	0.227	0.893
*Bifidobacterium*	0.165	3.626	0.598	0.638	−0.182	4.604	0.683	0.850	−0.336	4.247	0.670	0.893
*Blautia*	0.949	4.690	0.065	0.296	0.749	5.454	0.086	0.410	−0.123	3.729	0.831	0.943
*Clostridium*	−0.818	0.740	0.129	0.345	−0.455	0.480	0.514	0.822	0.492	0.768	0.394	0.893
*Collinsella*	0.012	0.020	0.947	0.947	0.528	0.018	0.413	0.735	0.461	0.019	0.477	0.893
*Coprococcus*	0.268	0.399	0.210	0.481	0.433	0.371	0.624	0.850	0.145	0.285	0.434	0.893
*Dorea*	−0.239	0.033	0.260	0.519	0.201	0.019	0.307	0.614	0.338	0.035	0.617	0.893
*Erysipelatoclostridium*	−0.414	0.627	0.510	0.582	−0.509	0.688	0.935	0.935	−0.083	0.770	0.943	0.943
*Eubacterium*	−0.607	0.029	0.077	0.296	−0.533	0.035	0.167	0.446	0.002	0.040	0.886	0.943
*Faecalibacterium*	−0.835	0.082	0.092	0.296	−1.024	0.054	0.048	0.381	0.192	0.090	0.940	0.943
*Gemmiger*	0.196	0.013	0.466	0.582	−0.535	0.213	0.744	0.850	−0.580	0.193	0.118	0.893
*Lachnoclostridium*	−0.738	0.257	0.353	0.565	−0.986	0.180	0.131	0.418	0.108	0.289	0.668	0.893
*Peptoclostridium*	1.034	0.050	0.078	0.296	0.887	0.057	0.305	0.614	−0.236	0.014	0.534	0.893
*Robinsoniella*	0.027	1.624	0.510	0.582	−0.747	1.938	0.102	0.410	−0.918	1.606	0.055	0.880
*Ruminococcus*	−0.502	0.331	0.468	0.582	0.145	0.192	0.744	0.850	0.637	0.316	0.433	0.893
*Subdoligranulum*	**−1.320**	**3.639**	**0.002**	**0.005**	**−0.997**	**2.025**	**0.006**	**0.022**	0.750	3.349	0.256	0.341

Differences are illustrated as effect size (E) expressed as the difference between groups in terms of the binary logarithm (2n), including the standard error (SE) of the effect size, *p* value, and false discovery rate (FDR) adjusted *p* value, which was considered significant at FDR < 0.05 (bold). NO, non-obese; MHO, metabolically healthy obesity; MUO, metabolically unhealthy obesity; E, effect size; SE, standard error; FDR, false discovery rate.

**Table 5 animals-14-02524-t005:** Pair-wise comparison (analysis of similarity, ANOSIM) of the microbial communities with three different dissimilarity measures; Bray–Curtis, unweighted, and weighted UniFrac.

Group	Dissimilarity	ANOSIM(R)	*p*-Value
NO VS. MHO	Bray–Curtis	0.061411	0.155
	weighted UniFrac	0.10961	0.068
	unweighted UniFrac	0.068168	0.177
NO VS. MUO	Bray–Curtis	0.034568	0.24
	weighted UniFrac	0.05679	0.168
	unweighted UniFrac	0.010425	0.393
NO VS. MUO	Bray–Curtis	−0.068446	0.826
	weighted UniFrac	−0.046296	0.713
	unweighted UniFrac	−0.055192	0.781

Healthy, non-obese (NO, *n* = 10), metabolically healthy obesity (MHO, *n* = 12), and metabolically unhealthy obesity (MUO, *n* = 8) groups.

## Data Availability

The original data presented in the study are openly available at https://doi.org/10.6084/m9.figshare.26058313.v2 (accessed on 23 June 2024).

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
