# Peer review of "Association between Gut Microbiota and Metabolic Health and Obesity Status in Cats"

_animals, 2024, doi:10.3390/ani14172524_

Round 1

Reviewer 1 Report

Comments and Suggestions for Authors

I found this paper to be well-written and interesting. I would have appreciated more background as to the basis for the classifications of MUO and MHO and the selection of cut off values. Also, and importantly, what characterizes the basis for definitions of "healthy" or "unhealthy" types. Yes one is likely statistically more "normal" than the other - but what about "health"? There are no associated data regarding well-being or development of disease or pathology.

I have some reservations about the microbiome methodologies employed (it is a complex and evolving field) and I am not an expert - but other reviewers may feel more confident about being specific and should be heeded. 

Nonetheless, the associations reported are interesting and merit publication. publication.  

Reviewer 2 Report

Comments and Suggestions for Authors

Thank you for the opportunity to review this nice paper. This is an interesting study on the gut microbiome of lean and obese cats, with a special look at obese cats metabolically unhealthy (OUM). The criteria for OUM classification agree with recent evidence in the field, and the study design is fine. I, particularly, do not have enough microbiome and microbiology knowledge to evaluate the related methods employed; however, the results reported are mostly in line with evidence in the human field, and previous studies on cats’ microbiomes. The discussion is very informative and properly explains the study’s findings.  I have a few comments to the authors that could be addressed before publication.

Line 24 – Hematological characteristics give an idea of analysis on parameters such as complete blood count and other related variables, please emphasize if this was hormone or serum biochemistry parameters.

Line 25 – Please introduce MHO and MUO in the abstract. Many researchers will going to read only this paragraph, and in this way, it should stand alone (including non-standard abbreviations)

Line 39 – The studies referenced do not support global cat overweight/obesity prevalence exceeding 60%. Ideally, authors would improve this section by bringing information from different epidemiologic studies that support cats’ overweight/obesity ranging from xx-xx% according to the region/country.

Line 42 – it is important to notice that obesity's impact on a cat’s life expectancy is yet to be demonstrated and that the information in this sentence refers to humans.

Line 72 – the BCS system and other morphometric measures seemed validated to help estimate the amount of body fat in a given individual, and there are several published materials in this line, including references 16 and 17 later cited in the paper.

MMs 2.1 – In my opinion, this section should just introduce how many cats were recruited informing the inclusion/exclusion criteria. However, information about how many were excluded and why, as well as how many were assigned to MHO or MUO groups should be displaced at the beginning of the results section.  This is important because it is stated that cats were excluded due to maladaptation to an intervention not previously explained.

Table 2 – Since BCS is not a continuum variable, rather, it is a categorical one, is better to express central data as median and not mean (i.e. 8.6 and 8.9 suggest mean and not median).

Line 303 – Maybe this paragraph could be suppressed or shortened according to the authors' and editors' feelings. The justificative for not including fSAA in this study is fine, but I do not feel discussing PCR is necessary since it was not evaluated in the study.

Line 320 – The assumption that Bifidobacterium-based probiotic supplementation could increase the risk of obesity should be treated cautiously. The study reported an association between the bacteria and the obesity state; however, a cause-consequence relationship can not be assumed by this study design. Please rephrase it. 

Reviewer 3 Report

Comments and Suggestions for Authors

Summary: This study aimed to characterise the gut microbiota and to determine whether alterations in the intestinal microbiota were associated with obesity and metabolic status in cats using 16S rRNA gene sequencing. Cats were divided into three groups: the first group consisted of cats with normal body weight (NO), and the other groups were composed of obese cats, one group metabolically healthy (MHO) and the third one metabolically unhealthy (MUB). The distinction between MHO and MUB cats was based on the BCS, BMI and adiponectin and triglyceride levels. Obese cats showed significantly higher levels of the Bifidobacteriaceae, Coriobacteriaceae, and Veillonellaceae families compared to non-obese cats and the abundance of the Ruminococcaceae family was higher in MUO cats compared to the other two groups. These changes were correlated with the BMI, and cholesterol and triglycerides levels, respectively.  

General and Specific comments

Thank you for conducting this study, obesity is the main nutritional disorder in cats.  As diet is one of the main factors that can influence gut microbiota and bacterial metabolites, it is valuable that you implement the same diet for all animals. Also, it is important that all the animals are neutered, as this is one of the main factors associated with obesity. However, the manuscript could benefit from some modifications.

The criteria for selecting metabolically healthy (MHO) and metabolically unhealthy obese (MUO) cats are not clear. The study you based your results upon, only had 10 cats of different ages, breeds and neutering status. Moreover, the cut-off values for each parameter were not based on this study. You established your cut-off values based on your healthy population, which was of small size as well. For example, values of adiponectin and SAA were quite different among studies. The cut-off value for Adiponectin in Okada’s study was 3 versus 1.53 in your study. For SAA, values were higher in NO cats. That means that if you had used those values most of your cats would have not been considered for the MOU group. In the general population, what would be considered the cut-off value?

Although it is known that the composition and function of the gut microbiome have been associated with obesity, are these changes considered contributing factors for obesity or are a consequence of it?  

How do you define obesity?

What are the limitations of the study besides the small sample size? Although changes in microbiome composition can be important, they do not reflect changes in microbiome function. There is a lot of redundancy in the gut microbiome. It is well known that the results of studies based on the 16S rRNA gene sequencing vary according to the sequencing and/or data analysis methods used. Many aspects of the methodology are missing. BCS does not always reflect visceral fat accumulation which is the one that is mainly implicated in metabolic derangements.

Specific comments

Line 39-42: this statement and the reference are for humans not for cats.

Line 84: what are the breeds of the cats? Was there a difference in activity levels, or any treats? differences in style of life?

Line 85-89: Numbers are confusing you said that initially 36 cats were included, then 3 cats were excluded for a total of 33 cats, but then the final population consists of 32 animals. Then you said seven males and 3 females in NO (10), 10 males and 2 females in MHO (12) and 5 males and 3 females in MUO (8) for a total of 30 cats.  Table 2: 31 animals.

Line 107: What is the nutritional composition of the diet and the main ingredients? Were cats fed ad libitum?

Line 117: Please describe the method of BMI calculation.

Line 134: How did you isolate DNA? Did you add controls? how did you perform the sequencing? what variable region of the 16S rRNA gene did you amplify? what were the conditions of the PCR? how did you check the quality and quantity of the DNA?

Line 139: why did you decide to use this setting? we are talking about faecal samples.

Line 144:  What value was used for rarefaction (the depth cutoff)?

Line. 149: in line 149 you said that used Bray Curtis for Anosim, but then in line 153 you said that also Unifrac. It is advisable to use qualitative as well as quantitative methods that also consider phylogeny.

Line 172: For statistical analysis, please consider the compositional nature of the data. Please specify versions of each program.

Line 194:  It is better to use one type of descriptive statistics, all medians and ranges. Moreover, it is not clear where you are using means and where you are using medians.

Figure 1:  it isn't easy to compare columns, and adjust values at 100%. Percentage values should cover 100%.

Line 140 - 247: Veillonellaceae belongs to the Negativicutes class, Coriobacteriaceae to the Coriobacteriales class, and in turn to Coriobacteriia. Bifidobacterium belongs to the Bifidobacteriacea family and actinomycetes to actinobacteria. It would be clearer to specify this.

Line 297: did you measure cholesterol levels? What were the values in each group? the same for bilirubin levels (line 281).

Line 298: What is the relationship between bilirubin levels and Ruminococcaceae?

Line 306: crucial point. 

Line 307: How useful is C reactive protein in cats?

Line 342:  How could these groups contribute to preventing obesity?

Line 397: Sequencing data should be available in a public repository like NCBI. Also, the formulas and methods used for statistical analysis.

Reviewer 4 Report

Comments and Suggestions for Authors

The authors presented a study linking the composition of the gut microbiome to the metabolic status of nutritional disorders (obesity) in cats. Studies of this type have already been performed in humans, but not in cats. The authors obtained a result indicating the dominant role of certain bacterial strains in the development and course of obesity. The paper is interesting and my critical comments are aimed at improving the scientific aspect of the work.

Specific comments:

Line 73 - in works of this type it is necessary to present the hypothesis (which is different from the aim of the work) and the methods used to verify this hypothesis.

Line 85 - it seems that age is crucial in the development of obesity. Unfortunately, nothing is known about the exact age of the individual cats used in the study as well as the overall veterinary status of the animals (e.g., whether they suffered from other diseases). Anyway, the range of 2-9 years seems to be too wide (this means that young and mature individuals were studied).

Line 120 - jugular veins are two: external and internal. Please clarify which one was used to draw blood.

Line 132 - the fact that the collection of the fecal sample depended on the owners strongly limits the reliability of the study. The question is whether an outsider can be 100% believed? Undoubtedly, such information should be marked as limitation of the study.
